# Ultrasound-Assisted Extraction of Carotenoids from Carrot Pomace and Their Optimization through Response Surface Methodology

**DOI:** 10.3390/molecules26226763

**Published:** 2021-11-09

**Authors:** Muhammad Umair, Saqib Jabbar, Mustapha M. Nasiru, Zhaoxin Lu, Jianhao Zhang, Muhammad Abid, Mian Anjum Murtaza, Marek Kieliszek, Liqing Zhao

**Affiliations:** 1Department of Food Science and Engineering, College of Chemistry and Engineering, Shenzhen University, Shenzhen 518060, China; umair_uaf@hotmail.com; 2Key Laboratory of Optoelectronic Devices and Systems, College of Physics and Optoelectronic Engineering, Ministry of Education and Guangdong Province, Shenzhen University, Shenzhen 518060, China; 3College of Food Science and Technology, Nanjing Agricultural University, Nanjing 210095, China; 2018208036@njau.edu.cn (M.M.N.); fmb@njau.edu.cn (Z.L.); nau_zjh@njau.edu.cn (J.Z.); 4Food Science Research Institute (FSRI), National Agricultural Research Centre (NARC), Islamabad 46000, Pakistan; Saqibjabbar2000@yahoo.com; 5Institute of Food and Nutritional Sciences, Pir Mehr Ali Shah, Arid Agriculture University Rawalpindi, Rawalpindi 44000, Pakistan; abidfoodscientist@yahoo.com; 6Institute of Food Science and Nutrition, University of Sargodha, Sargodha 40100, Pakistan; anjum.murtaza@uos.edu.pk; 7Department of Food Biotechnology and Microbiology, Institute of Food Sciences, Warsaw University of Life Sciences—SGGW, Nowoursynowska 159 C, 02-776 Warsaw, Poland

**Keywords:** carrot pomace, ultrasound assisted extraction, total carotenoids, β-carotene, response surface methodology

## Abstract

Ultrasound-assisted extraction (UAE) was used to extract carotenoids from the carrot pomace. To investigate the effect of independent variables on the UAE, the response surface methodology (RSM) with central-composite design (CCD) was employed. The study was conducted with three independent variables including extraction time (min), temperature (°C), and ethanol concentration (%). The results showed that the optimal conditions for UAE were achieved with an extraction time of 17 min, temperature of 32 °C, and ethanol concentration of 51% of total carotenoids (31.82 ± 0.55); extraction time of 16 min, temperature of 29 °C, and ethanol concentration of 59% for a combination of β-carotene (14.89 ± 0.40), lutein (5.77 ± 0.19), and lycopene (2.65 ± 0.12). The non-significant (*p* > 0.05) correlation under optimal extraction conditions between predicted and experimental values suggested that UAE is the more productive process than conventional techniques for the extraction of carotenoids from the carrot pomace.

## 1. Introduction

During the year 2010, the world total production of vegetables was approximately 1089 million tons and the carrot production was about 40 million tons. Among vegetables, the carrot (*Daucus carota*) is well known and China is the major carrot-producing country in the world [1]. The carrot is mainly consumed as raw or used to produce different products. In recent years, fruit and vegetable juices have become important due to the increase in overall demand for natural juice consumption instead of tea, coffee, and carbonated soft drinks [2,3]. Carrot juice is a good source of pro-vitamin A (carotene), vitamin B complex, and many minerals [4,5]. 

After processing vegetables in the food industry, a large quantity of waste is produced. According to one survey, only in Europe, approximately 27.94 million tons of food wastes are produced every year from the food processing industry [6]. The carrot pomace is also a by-product obtained after carrot juice processing. Only 60–70% juice yield is obtained after extraction of juice from carrots and up to 80% of carotene may be lost with remaining carrot pomace and until now; the leftover carrot pomace does not find any appropriate utilization [7,8]. However, carrot pomace contains a large amount of carotenoids, vitamins, dietary fiber, and minerals [9,10]. Carotenoids are the major precursor of vitamin A and help in the significant reduction in cancer, cardiovascular diseases and age-related macular degeneration [11,12]. Therefore, it is essential to extract carotenoids from carrot pomace.

Agricultural by-products produced during the handling and processing of fruits and vegetables, including cake, pomace, peels, seeds, leaves, bracts, cull fruits, and stones, represent a major waste disposal problem for the industry. Integrated utilization of food waste is a progressive direction of resource conservation. In almost every country in the world, the most important advances in scientific and technological progress and worldwide experience in the recycling of household and vegetable waste are used. Integrated use of food industry waste allows obtaining significant savings of material and energy resources, providing increased levels of closed production and resource cycles, which contributes to the economic efficiency of production. At the same time, the process of environmental pollution by waste is minimized. The integrated management of food industry waste is not only to use low-waste production technologies [13]. The involvement of wastes in chemical technology production processes as a secondary raw material makes it possible to turn them into a valuable product, that can be used in the chemical materials industry, pharmaceutical, and cosmetic industries. A significant amount of waste is generated during the processing of fruit and vegetable crops. Almost all of these wastes are secondary raw materials because they contain natural organic compounds. Therefore, the priority direction for the development of green chemical technologies is the search and production of organic compounds (plant extracts) from the waste of vegetable raw materials, as well as the study of their component composition and physicochemical properties of the obtained extracts, involvement in the production process of waste organic compounds obtained from waste of vegetable raw materials [14]. Nowadays, ultrasound-assisted extraction (UAE) is widely used for the extraction of nutritional material, such as bioactive compounds e.g., flavonoids [15,16,17], carotenoids [18], polysaccharides [19], proteins [20] and lipids [21]. Ultrasound extraction can enhance the extraction rate, extraction efficiency, reduce the extraction temperature and time as compared to traditional extraction methods [22]. The ultrasound instrument is not only simple in operation but also economically very cheap as compared to other extraction methods, such as microwave-assisted and supercritical fluid extractions. The main factor leading to the improvement of extraction yield during sonication is the ultrasonic cavitations because cavitations can cause locally high temperatures and pressures and free radicals [23,24] which may speed up or activate the chemical reactions of the extracted compounds. These radicals are mainly hydroxyl radicals that are generated when water is used as a solvent, and the formation of these free radicals depends on the dissolved gas species. The destruction of water molecules could produce highly reactive free radicals, which can modify other molecules, such as proteins [25].

If the input parameters or processes are not well optimized, efficient, and organized then only the use of low-cost materials cannot provide the desired output of removal efficiency. For optimization, the use of one factor at a time is non-feasible and also time-consuming. Another reason is that due to the lack of interactions among the factors, it is inadequate to obtain true optimum conditions. Due to this problem, nowadays, RSM is a generally used statistical tool for process optimization that can reduce cost, resources, and time [26,27]. Previously, most scientists extracted only polyphenols [15] or beta carotene from carrot pomace by sonication or other techniques as shown in Table 1 [28,29]. Therefore, this study was conducted with the aim to obtain the optimum conditions of different experimental factors, such as extraction temperature, time, and solvent concentration during ultrasonic extraction of carotenoids (β-carotene, lutein, and lycopene) from carrot pomace by using RSM with CCD. The results may not only be beneficial for the proper management of waste but also for the development of the carrot processing industry. 

## 2. Materials and Methods

### 2.1. Chemicals 

Acetonitrile was purchased from Sinopharm Chemical Regent Co., Ltd. (Shanghai, China). HPLC-grade methanol was obtained from Hanbon Science and Technology (Huaian, Jiangsu, China). Lutein and β-carotene were purchased from Sigma Aldrich Chemie GmbH (Steinheim, Germany). Sodium sulfate was obtained from Xilong Chemical Factory (Shantou, China). Butylated hydroxyl-toluene (BHT) was purchased from Sinopharm Chemical Regent Co., Ltd. (Shanghai, China). Acetone was obtained from Lingfeng Chemical Reagent Co., Ltd. (Shanghai, China). Petroleum ether and n-hexane were purchased from Nanjing Chemical Reagent Co., Ltd. (Nanjing, China). All other chemicals used were of analytical grade.

### 2.2. Carrot Pomace Powder Preparation 

Good quality carrots (*Daucus carota* L.) were purchased from the local market of Nanjing, China. Carrots were washed and sliced manually with a knife. After juice extraction by using a domestic juice extractor of MJ-M176P (Panasonic Manufacturing Berhad, Malaysia), the carrot pomace was obtained. A lab-scale freeze-dryer (Labconco Equipment Co. Kansas, MO, USA) was used for freeze-drying. A pressure (10 ± 5 Pa) and a temperature (−40 ± 1 °C) were automatically controlled. The raw material was kept in a freezer before freeze-drying at −20 °C for 24 h. On the dryer plates, the material was separated at a load of 4.0 kg m^−2^. The dried pomace after freeze-drying was grounded finely with a blade grinder and then stored in a desiccator until further use.

### 2.3. Optimization of Different Solvents 

The sonication technique was employed in order to extract carotenoids from carrot pomace powder by using the method of Jabbar et al. [15]. Preliminary studies were conducted on different solvents to determine optimum solvent concentration (25%, 50%, 75%), sample-to-solvent ratio (1 g/30 mL, 1 g/50 mL, and 1 g/70 mL) at a temperature (40 °C). After preliminary studies, five different solvents were used, including ethanol, methanol, acetone, acetonitirile, and n-hexane at 50% concentrations with 1 g/50 mL sample-to-solvent ratio at a temperature (40 °C). One gram of carrot pomace powder sample for each solvent was taken in 100 mL jacketed vessels separately in triplicate and 50 mL of solvent was added at 50% concentration. An ultrasonic processor of 750W (VC 750, Sonics and Materials Inc., Newtown, CT, USA) was used in this procedure (the detailed procedure is mentioned in Section 2.4). The yields (%) of the carotenoid extracts from carrot pomace powder by using different solvents were determined by the following formula: (1)$$Yield\;\left(\%\right)=\frac{Weight\;of\;the\;Extract}{Sample\;Weight}\times 100$$

The results of preliminary studies showed that ethanol showed the maximum yield of carotenoids compared to other solvents as shown in Figure 1. The variations in yield extractions using different solvents at different concentrations could be because of dissimilar polarities of the solvents used [38]. Therefore, we selected ethanol for further optimization of extraction time, temperature, and ethanol concentration during sonication by using response surface methodology. 

### 2.4. Ultrasound-Assisted Extraction (UAE) 

UAE of the carotenoids from carrot pomace was carried out in an ultrasonic processor of 750W (VC 750, Sonics and Materials Inc., Newtown, CT, USA) at a frequency and amplitude level of 20 kHz and 70%, respectively, by employing extraction parameters including: extraction time, temperature, and ethanol concentration. The determined ultrasonic intensity was 48 W cm^−2^ (Thermocouple HI 9063, Hanna Instruments Ltd., Leighton, Buzzard, UK). The dried ground pomace sample was placed in a 100 mL jacketed vessel at the ratio of 1 g/50 mL ethanol and then at different extraction conditions submitted to ultrasonic irradiation with a 0.5-inch probe by keeping 2 cm of its depth in the sample. The temperature was automatically controlled and when the set temperature was reached the sonication was started. A schematic diagram of the sonication system has been presented elsewhere [39]. After extraction, the filtration of the samples was performed with filter papers (Whatman No. 41 paper), and the extracts were centrifuged at 4000 rpm for 10 min. Then, evaporation was conducted at 30 °C to remove the solvent, and extracts were stored at 4 °C in micro-tubes until further use. For the determination of total carotenoids, lycopene, β-carotene, and lutein (method for determining the analytes is detailed below), the samples were re-suspended with distilled water at a concentration of 1 g L^−1^. All the measurements were performed in triplicate under the same conditions.

### 2.5. Experiment Design 

To determine the optimum solid/liquid ratio and amplitude level for the rest of the investigations, some preliminary experiments were performed. Therefore, 1 g/50 mL solid/liquid ratio and 70% amplitude level (ultrasonic intensity, 48 W cm^−2^) were selected for further studies. The RSM with CCD was used to optimize and investigate the effects of three independent variables such as extraction time (min, A), extraction temperature (°C, B), and solvent concentration (%, C) on total carotenoids, lycopene, β-carotene, and lutein. The low, mid, and high levels and their coded symbols for the CCD are shown in Table 2. Table 3 shows the complete experiment design of 17 combinations with three replicates at a central point. The second-order polynomial multiple regression equation was used to analyze the experimental data. The model equation was used as given below:(2)$$Y=\gamma_{0}+\sum_{i=0}^{n=3}\gamma_{i}X_{i}+\sum_{i=0}^{n=3}\gamma_{ii}X_{i}^{2}+\sum_{i=j=1}^{n=3}\gamma_{ij}X_{i}X_{j}\;$$
where *Y* is the predicted response, γ_0_ is the intercept, γ*_i_*,γ*_ij_*, and γ*_ii_* are the linear (main effect), interactive and quadratic model coefficients, respectively. Accordingly, *X_i_* and *X_j_* show the levels of the independent parameters and n is the number of factors analyzed. 

### 2.6. Determination of Total Carotenoids

The method of Liao et al. [40] was used for the determination of total carotenoids with slight modifications. The mixing of re-suspended extract sample (25 mL) and 80 mL of n-hexane/acetone (1:1, *v*/*v*) was carried out in a separation funnel and then held for 5 min. The organic phase was collected after separation and the aqueous phase was extracted repeatedly by using 15 mL of n-hexane/acetone (1:1, *v*/*v*) until it became colorless. The anhydrous sodium sulfate was added to the organic phase for dehydration. A spectrophotometer (Shanghai Jinghua Science and Technology Instruments Co., Ltd., Shanghai, China) was used to determine total carotenoids absorbance at 450 nm wavelength. The different β-carotene standard solution concentrations (2–10 µg/mL) were made. The results were expressed as µg β-carotene equivalent/g of sample. 

### 2.7. Determination of β-Carotene and Lutein

A method of Kim and Gerber [41] was used for the extraction of carotenoids from the re-suspended extract samples with some modification. Twenty-five milliliters of re-suspended extract sample was taken in a separation funnel and extracted three times with an equal volume of acetone and methanol. Then, the acetone–methanol extract was mixed vigorously with an equal volume of petroleum ether. The upper layer of petroleum ether was dehydrated by adding anhydrous sodium sulfate and after filtration, it was concentrated at 30 °C by using a rotary evaporator (Laborota 4000-efficient, Heidolph Instruments, Schwabach, Germany). Acetonitrile-methanol-acetone (40:40:20, *v*/*v*) solution was added to the concentrate and stored at −18 °C in the dark until further use. An Agilent 1100 series HPLC diode array detection (DAD) system with Agilent Zorbax Eclipse XDB-C18 Column (4.6 × 150 mm, with 5 µm particle size, USA) was used for the detection of β-carotene and lutein. The mobile phase consisted of acetonitrile-methanol-acetone (40:40:20, *v*/*v*), at a flow rate of 0.6 mL/min. Before injection into the column, the sample was filtered by using a syringe filter of 0.45 µm diameter and 20 µL injection volume was injected. The β-carotene and lutein were detected at 450 nm wavelength with retentions time of 5.614 and 12.395 min, respectively, and then calculated from a calibration curve against reference standards.

### 2.8. Determination of Lycopene

The contents of lycopene were measured by using a method described by Oliu et al. [42] with slight modifications. The re-suspended extract sample (0.6 mL) was mixed with 5 mL of BHT in acetone (0.05:99.95, *w*/*v*), 5 mL of ethanol (95:5, *v*/*v*), and 10 mL of n-hexane. The above mixture was centrifuged for 15 min at 320 g rpm. Then, 3 mL distilled water was added to it. The tube was then agitated for 5 min and held for 2 min at room temperature to allow phase separation. The absorbance of the upper n-hexane layer was determined by using a spectrophotometer at 503 nm against the blank. To calculate the lycopene contents the following equation was used:Lycopene = (∆503 × MW × DF × 1000)/(ε × L)(3)
where MW is the molecular weight of lycopene (536.9 g/mol), DF is the dilution factor, ε (172,000 L/mol cm) is the molar extinction coefficient for lycopene [39], and L is the path length in cm. The lycopene was expressed as µg/g of sample. All measurements were taken in triplicate.

### 2.9. Statistical Analysis

The experimental design and statistical analysis were carried out by using Design Expert 8.0.7.1 (Stat-Ease Inc., Minneapolis, MN, USA) software. To assess the goodness of fit of the regression models, the coefficients of determination (R2) and analysis of variance (ANOVA) were employed. The optimal extraction conditions of the three independent variables and each dependent variable were estimated by applying the three-dimensional RSM technique with CCD. All experiments were performed in triplicate.

## 3. Results and Discussion

### 3.1. Fitting the Response Surface Models

Table 2 shows the extraction yields of total carotenoids, β-carotene, lutein, and lycopene from the carrot pomace. By using the quadratic polynomial model, the second-order multiple regression Equation 1 was developed based on the results in Table 3. The analysis of variance (ANOVA), lack-of-fit, R-square values, and regression coefficients are shown in Table 4. The obtained results showed a good fit with the regression Equation 1 and the results were adequate with satisfactory R^2^ values and were statistically acceptable at different *p*-values. The ‘‘fitness’’ of the models was observed through the lack-of-fit tests (*p* > 0.1), which showed the feasibility of the models to predict the variations accurately (Prasad et al., 2011).

### 3.2. Influence of Independent Parameters on the Extraction of Total Carotenoids

The experimentally determined total carotenoids of the carrot pomace are shown in Table 3. It ranged from 11.53 to 32.63 μg/g. The response variable and the independent variables were related by applying multiple regression analysis on the experimental data. After neglecting non-significant terms, the following second-order polynomial equation was obtained:Y_1_ = −0.0786 + 0.1759A + 0.1203B + 0.0911C − 0.0012AB − 0.0033A^2^ − 0.0017B^2^ − 0.0009C^2^(4)

The analysis of variance of the quadratic regression model (Table 4) revealed that the values of the adjusted determination coefficient (adj. R^2^) and the determination coefficient (R^2^) were 0.9313 and 0.9699, respectively, which exhibited a high degree of correlation between the predicted and the observed values. It was observed that the model was significant (*p* < 0.001), and the lack of fit was non-significant at *p* > 0.1, indicating the model’s acceptability. The significance of each coefficient was evaluated by the *p*-values, which depict a linear interaction and quadratic pattern between the parameters. Table 4 shows that one linear (A) and two quadratic variables (A^2^ and B^2^) were significant at *p* < 0.001, two linear terms (B, C) were significant at *p* < 0.01, and one interaction (AB) was significant at *p* < 0.05 while one quadratic variable was highly significant at *p* < 0.0001. The results demonstrated that extraction time was the most significant parameter that affected the total carotenoids followed by extraction temperature and ethanol concentration.

The three-dimensional response surface plots are shown in Figure 2, which indicate the interactions between two variables by keeping the other variables at their fixed levels for total carotenoids extraction. The effects of interaction between extraction time and temperature on the total carotenoids are shown in Figure 2A. At a fixed extraction time, total carotenoids increased with the increase in extraction temperature from 10–35 °C, afterwards, total carotenoids decreased. The maximum total carotenoids recovery was obtained when the extraction time was around 20 min, afterwards, total carotenoids decreased with increasing extraction time. Similar trends were observed in the interaction effect between the other two factors on the total carotenoids (Figure 2B,C). When one factor was fixed, the other factor first increased and then decreased the total carotenoids, which is in agreement with the previous finding [43]. The change of extraction yield with temperature during sonication may be due to the combined effect of the cavitation and heat because with increasing temperature, the intensity of cavitation decreases [44,45] and also carotenoids are very sensitive to oxidation, isomerization or other thermal-triggered chemical reactions during processing because they have highly unsaturated compounds with an extensive conjugated double-bond system [46]. The extraction time also plays an important role in the extraction of total carotenoids because increasing the contact time of the solvent with solids may enhance the diffusion of the compounds [47]. Similarly, a proper ethanol concentration is also very important to take the maximum recovery of the carotenoids. It is probably due to the fact that in the aqueous solution the propagation of sonication waves increases but the use of a solvent with a high amount of water > 60% can result in increased radicals’ production from the ultrasound dissociation of water. Due to these high-energy species, the oxidative reactions can cohabit with the extraction reactions, and therefore, the extraction efficiency of the target compounds decreases [48]. Generally, ultrasound can enhance extraction because ultrasound waves can rupture the cell walls by penetrating into the matrix material [49], resulting in carotenoids being more easily discharged from the matrix into the extraction medium. Moreover, ultrasound can enhance the solvent extraction power and extract the targeted components by driving the solvent into the matrix [43].

### 3.3. Influence of Independent Variables on the Extraction of Different Carotenoids

The chemical structures of three different carotenoids are depicted in Figure 3. It is shown that due to the different chemical properties, it is difficult to develop a single process for the optimal extraction of carotenoids.

#### 3.3.1. β-Carotene

The results regarding the effect of independent variables on the extraction of β-carotene are presented in Table 3. The minimum and maximum extraction yield of β-carotene were obtained in run 13 (5.24 μg/g) and run 14 (14.80 μg/g), respectively. The final predictive equation obtained for β-carotene by using the significant terms is described as below:Y_2_ = 1.1602 + 0.0606A + 0.0498B + 0.0605C − 0.0006AB − 0.0013A^2^ − 0.0008B^2^ − 0.0006C^2^(5)

For β-carotene, the two independent variables were highly significant (*p* < 0.0001) in two linear (A, B) and two quadratic terms (B^2^, C^2^) while one linear (C) and one quadratic term (A^2^) was significant at *p* < 0.001 and one interaction term (AB) was significant at *p* < 0.01 (Table 4).

The three-dimensional response surface plots (Figure 4) show the effect of independent variables on the extraction efficiency of β-carotene. Figure 4A indicates that the extraction efficiency was highly affected by the extraction time and temperature. It may be attributed to the combination of thermal and cavitation effects but at a certain temperature and time the effect of cavitation intensity decreases, and hence it affects the final extraction yield [50]. Figure 4B shows that the linear increase in extraction time (3–20 min) at a fixed ethanol concentration led to a marked increase in the extraction yield of β-carotene and a linear increase in ethanol concentration (13–55%) at a fixed extraction time also led to a marked increase in β-carotene content. Figure 4C shows the interaction of the extraction temperature and ethanol concentration. It was found that the maximum extraction yield was achieved when the extraction temperature was 35 °C and the ethanol concentration was 55%.

#### 3.3.2. Lutein

Table 3 shows the effect of independent parameters on the extraction yield of lutein. The highest extraction yield was obtained in run 2 (5.41 μg/g) and the lowest extraction yield was obtained in run 12 (3.36 μg/g). The final second-order polynomial equation obtained by neglecting the non-significant terms is given below:Y_3_ = 1.2015 + 0.0294A + 0.0279B + 0.0128C − 0.0003AC − 0.0006A^2^ − 0.0004B^2^ − 0.0001C^2^(6)

The linear, interaction, and quadratic effects of each independent variable on lutein contents are shown in Table 4. The results showed that all the linear variables (A, B, C) and one quadratic variable (B^2^) were significant at *p* < 0.0001, whereas one interaction variable (AC) and one quadratic variable (C^2^) were significant at *p* < 0.01, while one quadratic variable (A^2^) was significant at *p* < 0.001.

Figure 5 shows the three-dimensional response surface effects of independent variables on the extraction efficiency of lutein. Figure 5A shows that the extraction temperature and time significantly affected the extraction yield of the lutein contents. The extraction time and temperature showed maximum extraction efficiency at a certain level, but a further increase resulted in the degradation of the lutein contents. Figure 5B shows that at a fixed temperature, the increase in extraction time and ethanol concentration led to a marked increase in the extraction of lutein contents at certain levels, afterward, a decreasing trend was observed. Figure 5C shows the interaction of the extraction temperature and ethanol concentration. It was observed that the optimal extraction yield was obtained when the ethanol concentration was 80% and the extraction temperature was 20 °C.

#### 3.3.3. Lycopene

The effects of independent parameters on the extraction yield of lycopene are shown in Table 3. The highest and lowest values were obtained in run 2 (2.50 μg/g) and in run 12 (0.46 μg/g), respectively. The final second-order polynomial equation obtained by using the significant terms is described below:Y_4_ = −0.2010 + 0.0431A + 0.0494B + 0.0205C − 0.0004AC − 0.0010A^2^ − 0.0007B^2^ − 0.0001C^2^(7)

Table 4 shows the linear, interaction, and quadratic effects of each independent variable on lycopene extraction yield. The results showed that two linear variables (B, C) and one quadratic variable (B^2^) were significant at *p* < 0.0001, one linear (A) and one quadratic variable (A^2^) was significant at *p* < 0.001 whereas one interaction variable (AC) was significant at *p* < 0.05 and one quadratic variable (C^2^) was significant at *p* < 0.01.

The three-dimensional contour plots demonstrated the effects of three independent variables on the extraction yield of lycopene are shown in Figure 6. Figure 6A indicates that the lycopene extraction yield increased with an increase in extraction temperature (10–20 °C) and extraction time (3–10 min). Afterwards, a decreasing trend was observed in the extraction yield of the lycopene; similar results were observed in a previous study [51]. Oxidation and isomerization are the main reasons for lycopene degradation during thermal processing [52], resulting in fragment products such as methyl-heptenone, acetone, glyoxal, and laevulinic aldehyde [53,54]. Generally, the observed changes during ultrasonic extraction of lycopene may be caused by cavitation, which regulates various physical, chemical, and biological reactions [55]. Figure 6B,C demonstrate that 80% ethanol concentration resulted in the maximum extraction yield of lycopene.

### 3.4. Predictive Models Verification

For total carotenoids and a combination of three different carotenoids, the estimated levels of optimum extraction conditions are described in Table 5. The predicted extraction conditions for total carotenoids were 17 min extraction time, 32 °C extraction temperature, and 51% ethanol concentration for the maximum extraction of total carotenoids (32.20 μg/g). Moreover, for the combination of three different carotenoids such as β-carotene (14.37 μg/g), lutein (5.35 μg/g), and lycopene (2.50 μg/g), the predicted extraction conditions were 16 min extraction time, 29 °C extraction temperature, and 59% ethanol concentration. To compare the predicted results with the experimental values, the experiment re-checking was performed by using the optimum extraction conditions for each response. Mean values of 31.82 μg/g total carotenoids, 14.89 μg/g β-carotene, 5.77 μg/g lutein, and 2.65 μg/g lycopene acquired from actual experiments confirmed the acceptability of the RSM models. The results of total carotenoids and three different carotenoids show that there was no significant difference (*p* > 0.05) between predicted and experimental values. Consequently, the models can be utilized to optimize the extraction conditions of carotenoids for carrot pomace.

## 4. Conclusions

The present study indicated that the carrot pomace is a rich source of carotenoids and the use of ultrasound was a productive method for the extraction of carotenoids from the carrot pomace since it could greatly decrease the extraction time as compared to other extraction methods. The RSM with CCD was successfully employed to optimize the extraction conditions of carotenoids from carrot pomace. A remarkable effect on the extraction yields of all responses was observed by the independent variables such as time, temperature, and ethanol concentration. The optimum extraction conditions were obtained for the total carotenoids, β-carotene, lutein, and lycopene. In conclusion, ultrasound-assisted extraction has the potential for the extraction of carotenoids from carrot pomace.

## Figures and Tables

**Figure 1 molecules-26-06763-f001:**
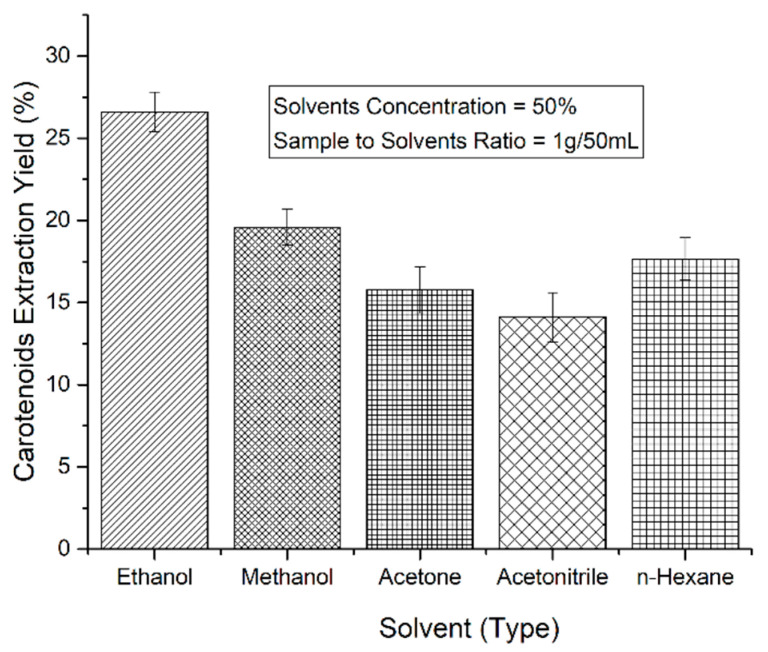
Yield (%) of carotenoids extracts from carrot pomace powder by using different solvents.

**Figure 2 molecules-26-06763-f002:**
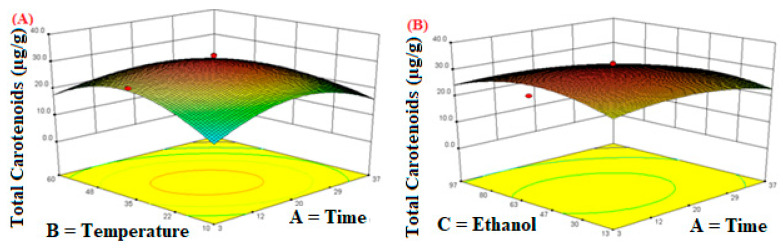
The response surface plots of the total carotenoids as influenced by independent variables during extraction. (**A**) Time; (**B**) Temperature; (**C**) Ethanol.

**Figure 3 molecules-26-06763-f003:**
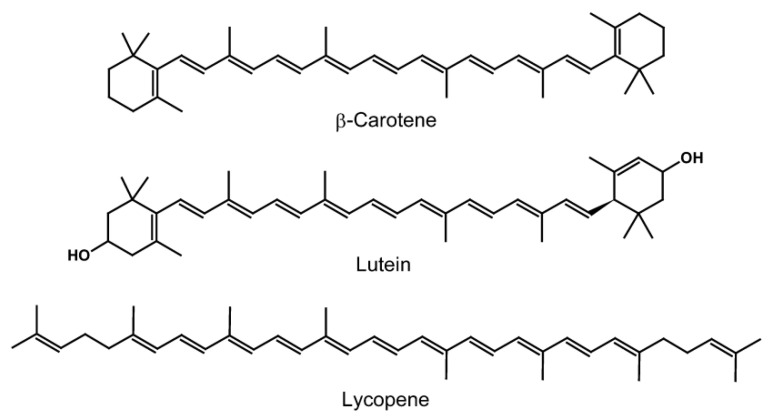
Chemical structures of β-carotene, lutein, and lycopene.

**Figure 4 molecules-26-06763-f004:**
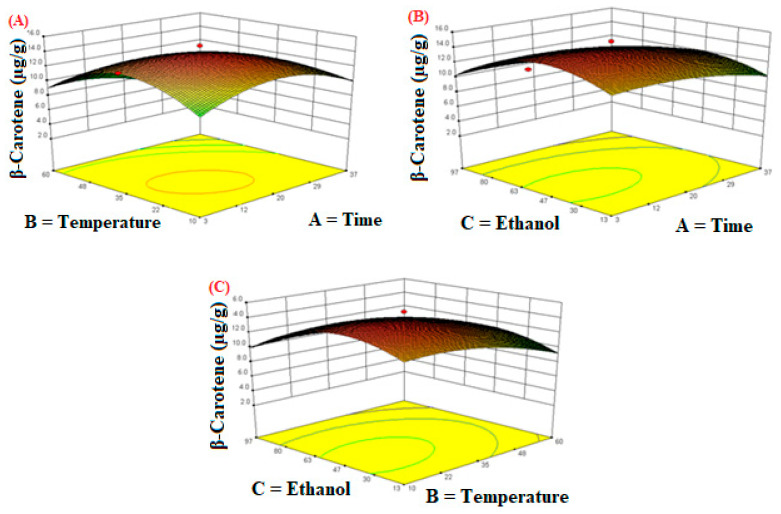
The response surface plots of the β-carotene as influenced by independent variables during extraction. (**A**) Time; (**B**) Temperature; (**C**) Ethanol.

**Figure 5 molecules-26-06763-f005:**
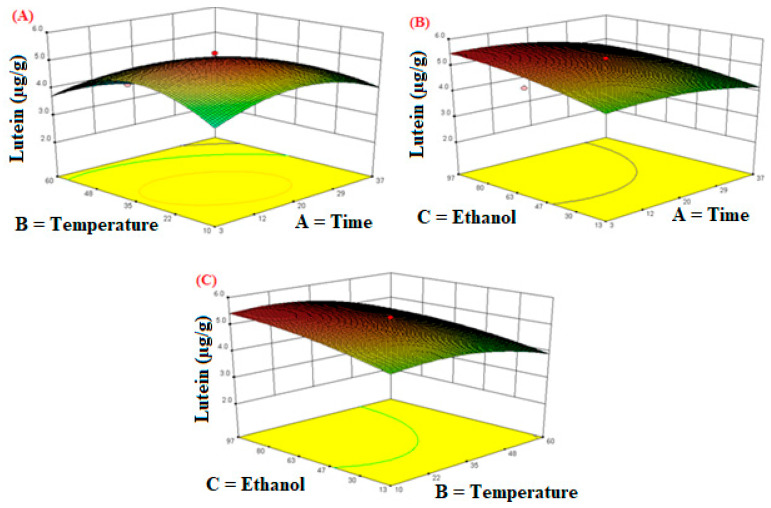
The response surface plots of the lutein as influenced by independent variables during extraction. (**A**) Time; (**B**) Temperature; (**C**) Ethanol.

**Figure 6 molecules-26-06763-f006:**
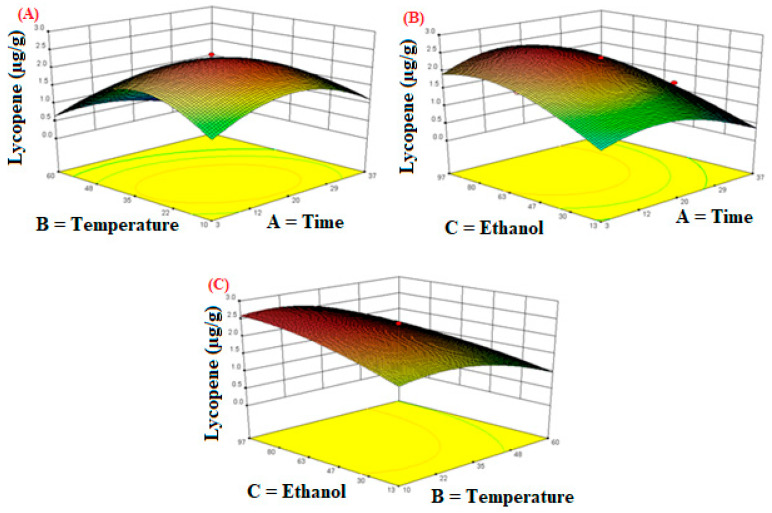
The response surface plots of the lycopene as influenced by independent variables during extraction. (**A**) Time; (**B**) Temperature; (**C**) Ethanol.

**Table 1 molecules-26-06763-t001:** Application of different extraction technologies in the recovery of carotenoids.

Extraction Method	Waste Material	Compounds Recovered	Conclusion	References
Water-induced hydrocolloidal complexation for extraction	Carrot Peel	β-carotene	The adaptability of the carotene–pectin hydrocolloidal complexation in the extraction of carotene from carrot peel waste was proven to be successful. The complexation process requires no organic solvent and relies on water addition to induce the formation of hydrocolloidal system. The purity of b-carotene fractionated from the complex is identical to the b-carotene extracted using solvent extraction, which was 96%.	[30]
Microwave-assisted extraction	Carrot pomace and peel	Total carotenoids, β-carotene content	A 77.48% recovery of carotenoid was achieved successfully at optimum conditions (165 W of microwave power, 9.39 min of extraction time, and 8.06:1 g/g of oil-to-waste ratio); hence the carotenoid extraction by using oil under microwave irradiation is a promising process. The use of intermittent microwave radiation to enhance the MAE of b-carotene and carotenoids from carrot peels was investigated. Combined use of lower microwave power (180 W) and solvent volume (75 mL) or higher microwave power (300 W) and solvent volume (150 mL) along with a lower intermittency ratio (a = 1/4) resulted in higher contents of b-carotene and total carotenoids of the extracts	[31,32]
Electrohydrodynamic-ultrasonic procedure for extraction	Carrot pomace	β-carotene	In this research, the influence of the EHD process before the ultrasonic process for β-carotene extraction from carrot pomace powder was investigated. The results showed that increasing the EHD time from 2.5 to 20 min increased the β-carotene concentration.	[28]
Ultrasound treatment	Carrot slice	Total carotenoids	The changes in carrot tissue caused by ultrasound treatment had an impact on total carotenoid content and color changing. Ultrasonic treatment, especially in the case of using ultrasound at 35 kHz, resulted in a substantial increase in carotenoids content in comparison to raw carrot, which was probably related to the destruction of the original cellular structure and could facilitate the extraction of these compounds.	[33]
Supercritical CO_2_ extraction process	Carrot peel	Total carotenoids	This work aimed to assess and optimize the extraction of carotenoids from carrot peels by supercritical CO_2_ (SCO_2_), utilizing ethanol as a co-solvent. The evaluated variables were temperature, pressure and co-solvent concentration. According to the validated model, the optimal conditions for maximum mass yield (5.31%, d.b.) were found at 58.5 °C, 306 bar, and 14.3% of ethanol, and at 59.0 °C, 349 bar, and 15.5% ethanol for carotenoid recovery (86.1%).	[34]
Pulsed electric field	Carrot puree Carrot pomace Tomato peel	Total carotenoids, β-carotene, lycopene	This study shows the feasibility of using PEF treatment to develop functional natural food ingredients, for example, carrot pomace with improved carotenoid extractability. Electroporation due to PEF treatment can be used to improve the extractability of carotenoids in carrot pomace with limited loss of carotenoids into the juice during extraction. The suitable extraction conditions were obtained at extraction time 49.4 min, extraction temperature 52.2 °C, and extraction ratio 1:70 (*w*/*w*). Under these conditions, the response variables were predicted to be 19.6 μg/g, 0.27, and 74 nm for β-carotene content. The results of this work demonstrated that the application of PEF pre-treatment of moderate intensity (5 kV/cm) and relatively low energy input (5 kJ/kg) before solvent extraction process with either acetone or ethyl lactate, can represent a sustainable, environmentally friendly, and food safety approach to intensify the extractability of carotenoids, especially lycopene, from industrial tomato peels residues.	[35,36,37]

**Table 2 molecules-26-06763-t002:** Levels of independent variables of the experimental design.

Symbols	Independent Parameters	Units	Low Level	Mid Level	High Level
A	Time	Min	03	20	37
B	Temperature	°C	10	35	60
C	Ethanol	%	13	55	97

**Table 3 molecules-26-06763-t003:** Central-composite design (un-coded) for extraction of total carotenoids, β-carotene, lutein, and lycopene from carrot pomace (μg/g).

Run	A (Min)	B (°C)	C (%)	Total Carotenoids	β-Carotene	Lutein	Lycopene
1	3.00	35.00	55.00	26.60	13.20	4.77	1.87
2	10.00	20.00	80.00	19.58	9.57	5.41	2.50
3	20.00	35.00	55.00	32.63	13.79	5.27	2.38
4	10.00	50.00	30.00	19.67	9.65	4.00	1.10
5	10.00	20.00	30.00	22.53	12.26	4.44	1.54
6	20.00	10.00	55.00	24.52	12.29	4.61	1.71
7	20.00	35.00	13.00	20.90	8.23	4.06	1.17
8	20.00	35.00	55.00	29.34	13.57	5.14	2.25
9	37.00	35.00	55.00	18.78	9.30	4.12	1.22
10	30.00	20.00	80.00	17.65	9.19	5.01	2.11
11	30.00	20.00	30.00	21.79	11.32	4.25	1.36
12	30.00	50.00	30.00	15.76	7.28	3.36	0.46
13	20.00	35.00	97.00	14.11	5.24	5.29	2.40
14	20.00	35.00	55.00	32.14	14.80	5.05	2.16
15	30.00	50.00	80.00	11.53	6.16	3.75	0.85
16	20.00	60.00	55.00	18.10	8.20	3.60	0.70
17	10.00	50.00	80.00	22.07	9.02	4.86	1.46

**Table 4 molecules-26-06763-t004:** Results of analysis of variance and regression coefficients for total carotenoids, β-carotene, lutein, and lycopene.

Source	Total Carotenoids	β-Carotene	Lutein	Lycopene
γ_0_	−0.0786 ^b^	1.1602 ^a^	1.2015 ^a^	−0.2010 ^a^
A	0.1759 ^b^	0.0606 ^a^	0.0294 ^a^	0.0431 ^b^
B	0.1203 ^c^	0.0498 ^a^	0.0279 ^a^	0.0494 ^a^
C	0.0911 ^c^	0.0605 ^b^	0.0128 ^a^	0.0205 ^a^
AB	−0.0012 ^d^	−0.0006 ^c^	−0.0003 ^c^	−0.0004 ^d^
AC	−0.0005 ^NSa^	0.0000 ^NSb^	−0.0001 ^NSb^	0.0000 ^NSb^
BC	0.0002 ^NSb^	0.0001 ^NSa^	0.0000 ^NSb^	−0.0001 ^NSb^
A^2^	−0.0033 ^b^	−0.0013 ^b^	−0.0006 ^b^	−0.0010 ^b^
B^2^	−0.0017 ^b^	−0.0008 ^a^	−0.0004 ^a^	−0.0007 ^a^
C^2^	−0.0009 ^a^	−0.0006 ^a^	−0.0001 ^c^	−0.0001 ^c^
*p*-Value	0.0002	<0.0001	<0.0001	<0.0001
*F*-Value	25.08	68.51	54.93	39.76
R^2^	0.9699	0.9888	0.9860	0.9808
Adj. R^2^	0.9313	0.9743	0.9681	0.9561
Lack-of-fit	0.5306 ^NSb^	0.7477 ^NSb^	0.4764 ^NSb^	0.2664 ^NSb^

^a^ Significant at *p* < 0.0001, ^b^ Significant at *p* < 0.001, ^c^ Significant at *p* < 0.01, ^d^ Significant at *p* < 0.05, ^NSa^ Non-Significant at *p* > 0.05, ^NSb^ Non-Significant at *p* > 0.1, adj. R^2^: Adjusted R^2^.

**Table 5 molecules-26-06763-t005:** Predicted and experimental values of total carotenoids, β-carotene, lutein, and lycopene under the optimal extraction conditions (μg/g).

Response Variables	Optimum Extraction Conditions			Maximum Value (μg/g)	
	Time (Min)	Temp (°C)	Ethanol (%)	Predicted	Experimental ^a^
Total carotenoids	17	32	51	32.20	31.82 ± 0.55
β-carotene	16	29	59	14.37	14.89 ± 0.40
Lutein	16	29	59	5.35	5.77 ± 0.19
Lycopene	16	29	59	2.50	2.65 ± 0.12

^a^ Means ± standard deviation (n = 3).

## Data Availability

Not applicable.

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
