# Peer review of "Ultrasound-Assisted Extraction of Carotenoids from Carrot Pomace and Their Optimization through Response Surface Methodology"

_molecules, 2021, doi:10.3390/molecules26226763_

Round 1

Reviewer 1 Report

  1. The title of the manuscript is very general, and it needs to reflect the content. Please mention the name of the food waste and the compounds extracted in the title.
  2. The quantitative amounts of extracted compounds should be mentioned in the abstract.
  3. The way citations have been cited in the introduction is inappropriate. After each statement, cite the paper from where that statement has been taken. Citing five or six papers every time is not good.
  4. Why dosage of the food waste and parameters related to sonication (intensity, frequency, power) itself were not optimized? They may affect the extraction yield. Please include.
  5. Include the optimization of solvent type.
  6. Please compare the efficiency of the UAE with the traditional and other advanced methods reported in the literature for extraction of selected compounds from carrot waste. Present in form of the table.
  7. Please highlight the difference between this paper and the one published earlier on the extraction of phenolic compounds from carrot waste. Discuss that paper in the introduction.
  8. How did you deal with the waste generated after the UAE of carrot pomace?
  9. How the extracted compounds can be employed in practical life?

Author Response

List of Changes

Manuscript ID: Molecules-1438414

Title: “Ultrasound-assisted extraction of carotenoids from carrot pomace and their optimization through response surface methodology”

Journal: Molecules

Referee #1

  1. The title of the manuscript is very general, and it needs to reflect the content. Please mention the name of the food waste and the compounds extracted in the title.

Line 1: The comment followed and title has been changed according to your valuable comment. “Ultrasound-assisted extraction of carotenoids from carrot pomace and their optimization through response surface methodology”.

  1. The quantitative amounts of extracted compounds should be mentioned in the abstract.

Line 28-30: The comment followed and quantitative amounts have been added into revised manuscript.  “Total carotenoids (31.82 ± 0.55), extraction time of 16 min, temperature of 29 ℃ and ethanol concentration of 59% for a combination of β-carotene (14.89 ± 0.40), lutein (5.77 ± 0.19) and lycopene (2.65 ± 0.12)”.

  1. The way citations have been cited in the introduction is inappropriate. After each statement, cite the paper from where that statement has been taken. Citing five or six papers every time is not good.

Line 42-102: The references have been arranged according to your valuable suggestion in the revised manuscript.

  1. Why dosage of the food waste and parameters related to sonication (intensity, frequency, power) itself were not optimized? They may affect the extraction yield. Please include.

Previous studies showed that frequency 20 kHz, amplitude level of 70% and ultra-sonic intensity 48 W cm-2 give maximum extraction. That’s why in our previous studies, we have already mentioned it. The current study is the extension of our previous study.

Jabbar, S., Abid, M., Wu, T., Hashim, M. M., Saeeduddin, M., Hu, B., & Zeng, X. (2015). Ultrasound‐assisted extraction of bioactive compounds and antioxidants from carrot pomace: A response surface approach. Journal of Food Processing and Preservation, 39(6), 1878-1888.

  1. Include the optimization of solvent type.

In the current study, we have only selected one solvent i.e. ethanol. The aim of our current study is to optimized different concentrations of ethanol for maximum recovery of carotenoids as shown in Table 2 and Table 4. In the future we will work on comparison with other solvents. Thanks for your valuable comment.

  1. Please compare the efficiency of the UAE with the traditional and other advanced methods reported in the literature for extraction of selected compounds from carrot waste. Present in form of the table.

In the current study, we have only optimized sonication conditions for maximum extraction of carotenoids. In the future we will work on comparison with other methods. Thanks for your valuable comment.

  1. Please highlight the difference between this paper and the one published earlier on the extraction of phenolic compounds from carrot waste. Discuss that paper in the introduction.

Line 100-102: The comment followed. We have added in the revised manuscript. Thanks for your valuable comment. “Previously, mostly scientists extract only polyphenols (Jabbar et al., 2015) or beta carotene form carrot pomace by sonication or other techniques (Salehi & Dinani, 2020; Tiwari et al., 2019)”.

  1. How did you deal with the waste generated after the UAE of carrot pomace?

Actually, it was lab scale experiment that’s why we bought fresh carrot from the market  according to experiment design. Then after juice extraction, the pomace was dried in freeze drier and then dried powder was used during extraction. After carotenoids extraction the remaining waste material was dumped into the soil.

  1. How the extracted compounds can be employed in practical life?

We can use extracted compounds into different fruit bars as we have recently published one research paper on it. The reference is given below:

Safdar, M. N., Kausar, T., Nadeem, M., Murtaza, M., Sohail, S., Mumtaz, A., Siddiqui, N., Jabbar, S., & Afzal, S. (2021). Extraction of phenolic compounds from (Mangifera indica L.) and kinnow (Citrus reticulate L.) peels for the development of functional fruit bars. Food Science and Technology.  https://doi.org/10.1590/fst.09321.

Reviewer 2 Report

This is an interesting manuscript dedicated to the use of ultrasound-assisted extraction (UAE) to recover carotenoids from the carrot pomace.

My first comment is that authors must decide which citation style they want to follow, as the citations are in Author-Date style but the papers in the references section are sorted by numbers. The manuscript also needs to be revised by a native English speaker.

Some particular comments:

Line 28 – The non-significant (p > 0.05) correlation under optimal extraction conditions between predicted and experimental values suggested that UAE is more productive process for the extraction of carotenoids from the carrot pomace

Please correct to …suggested that UAE is THE more productive process…

Line 63“…vegetable pomace has also become a source of pollution to the environment.”

The sentence is correct but authors need to include some supporting references.

Additionally, it is not entirely obvious the link between this sentence and the following “…to decrease this problem, new extraction technologies are needed”.

This is not the only problem that justifies the development of more productive industrial processes on extracting carotenoids from the carrot pomace. Authors should clearly present the economic impacts (to decrease industrial wastes and improve profits), environmental (green impacts by decreasing environmental disposal) and health impacts. Please reformulate this sentence.

Line 76 – “The main factor leading to the improvement of extraction during sonication is the ultrasonic cavitations because cavitations can cause locally high temperatures and pressures, and free radicals which may speed up or activate the chemical reactions of the extracted compounds.”

I think this sentence has several imprecisions:

  1. improvement of extraction” should be “improvement of extraction yield
  2. “cavitations can … free radicals which may speed up or activate the chemical reactions of the extracted compounds” – what reactions? Please be more specific.

Line 86 – “RSM is generally used statistical tool for process optimization that can reduce cost, resources and time

RSM is a generally…

Line 166 – “Twenty five milliliter re-suspended extract sample”

Please change to “Twenty five milliliters of”

Line 185 – “The mixture was centrifuged at 320 g for 15 min. And after shaking distilled water (3 mL) was added in it”

Please reformulate the sentence.

Please compare two sentences:

Line 249 vs Line 79

“The main factor leading to the improvement of extraction during sonication is the ultrasonic cavitations because cavitations can cause locally high temperatures and pressures…”

 Vs

…carotenoids are very sensitive to oxidation, isomerization or other thermal-triggered chemical reactions during processing because they have highly unsaturated compounds with an extensive conjugated double-bond system”

My question is: if carotenoids are sensitive to oxidation, isomerization or other thermal-triggered chemical reactions, and ultrasound cavitation causes locally high temperatures and pressures, how can be considered their use on an industrial basis, where the larger scales makes difficult to control the parameters?

Please comment.

  1. Results and discussion

The structure of this section does not seem appropriate as 3.4 (β-carotene), 3.5 (Lutein) and 3.6 (Lycopene) should be subsections of 3.3 - Influence of independent variables on the extraction of different carotenoids.

Additionally, it is not clear why the response surface plots of the Total Carotenoids and β-carotene are grouped in a same figure, which difficult the discussion. The same happens with figure 3 that groups the response surface plots for Lutein and Lycopene.

Line 361 – “…lutein and lycopene from the carrot pomace. Because it could greatly…

Please reformulate

Line 365 – “…The two most productive optimum extraction conditions were obtained…”

Authors must choose: “Most productive” or “optimum”.

I am confused. Which are the “two most productive” conditions?

Line 367 – “In conclusion, the ultrasound-assisted extraction may successfully be employed in the industry for the extraction of carotenoids from the carrot pomace.”

Probably this sentence is overambitious as this is a laboratorial study and not a pilot plant study.

Author Response

List of Changes

Manuscript ID: Molecules-1438414

Title: “Ultrasound-assisted extraction of carotenoids from carrot pomace and their optimization through response surface methodology”

Journal: Molecules

Referee #2

Some particular comments:

Line 28 – The non-significant (p > 0.05) correlation under optimal extraction conditions between predicted and experimental values suggested that UAE is more productive process for the extraction of carotenoids from the carrot pomace

Please correct to …suggested that UAE is THE more productive process…

Line 31-32: The comment has been followed. “UAE is the more productive process than conventional techniques”

Line 63 – “…vegetable pomace has also become a source of pollution to the environment.”

The sentence is correct but authors need to include some supporting references.

Additionally, it is not entirely obvious the link between this sentence and the following “…to decrease this problem, new extraction technologies are needed”.

This is not the only problem that justifies the development of more productive industrial processes on extracting carotenoids from the carrot pomace. Authors should clearly present the economic impacts (to decrease industrial wastes and improve profits), environmental (green impacts by decreasing environmental disposal) and health impacts. Please reformulate this sentence.

Line 58-78: The comment followed and new paragraph has been added in the revised manuscript. “Agricultural by-products produced during handling and processing of fruits and vegetables, including cake, pomace, peels, seeds, leaves, bracts, cull fruits and stones, represent a major waste disposal problem for the industry. Integrated utilization of food waste is a progressive direction of resource conservation. In almost every country in the world, the most important advances in scientific and technological progress and worldwide experience in the recycling of household and vegetable waste are used. Integrated use of food industry waste allows obtaining significant savings of material and energy resources, providing increased levels of closed production and resource cycles, which contributes to the economic efficiency of production. At the same time, the process of environmental pollution by the waste is minimized. The integrated management of food industry waste is not only to use low-waste production technologies (Vorobyova, V., Skiba, M., Miliar, Y., & Frolenkova, S. 2021). The involvement of wastes in chemical technology production processes as a secondary raw material makes possible to turn them into a valuable product, that can be used in the chemical materials industry, pharmaceutical and cosmetic industries. A significant amount of waste is generated during the processing of fruit and vegetable crops. Almost all of these wastes are secondary raw materials because they contain natural organic compounds. Therefore, the priority direction for the development of green chemical technologies is the search and production of organic compounds (plant extracts) from the waste of vegetable raw materials, as well as the study of their component composition and physicochemical properties of the obtained extracts, involvement in the production process of waste organic compounds obtained from waste of vegetable raw materials (Picot-Allain, Mahomoodally, Ak, & Zengin, 2021)”.

Line 76 – “The main factor leading to the improvement of extraction during sonication is the ultrasonic cavitations because cavitations can cause locally high temperatures and pressures, and free radicals which may speed up or activate the chemical reactions of the extracted compounds.”

I think this sentence has several imprecisions:

  1. improvement of extraction” should be “improvement of extraction yield

Line 86-87: The comment followed. “The main factor leading to the improvement of extraction yield”.

  1. “cavitations can … free radicals which may speed up or activate the chemical reactions of the extracted compounds” – what reactions? Please be more specific.

Line 90-93: The comment followed. “These radicals are mainly hydroxyl radicals that are generated when water is used as a solvent, and the formation of these free radicals depends on the dissolved gas species. The destruction of water molecules could produce highly reactive free radicals, which can modify other molecules, such as proteins (Wen et al., 2018).”

Line 86 – “RSM is generally used statistical tool for process optimization that can reduce cost, resources and time 

RSM is a generally…

Line 98-101: The comment followed. “RSM is a generally used statistical tool for process optimization that can reduce cost, resources and time (Gundupalli & Bhattacharyya, 2021; Pais-Chanfrau et al., 2021)”.

Line 166 – “Twenty five milliliter re-suspended extract sample”

Please change to “Twenty five milliliters of”

Line 171: The comment followed. “Twenty five milliliters of re-suspended extract sample”

Line 185 – “The mixture was centrifuged at 320 g for 15 min. And after shaking distilled water (3 mL) was added in it”

Please reformulate the sentence.

Line 189-190: The comment followed. “The above mixture was centrifuged for 15 minutes at 320 g rpm. Then 3 mL distilled water was added in it.”

Please compare two sentences:

Line 249 vs Line 79

 “The main factor leading to the improvement of extraction during sonication is the ultrasonic cavitations because cavitations can cause locally high temperatures and pressures…”

 Vs

…carotenoids are very sensitive to oxidation, isomerization or other thermal-triggered chemical reactions during processing because they have highly unsaturated compounds with an extensive conjugated double-bond system”

My question is: if carotenoids are sensitive to oxidation, isomerization or other thermal-triggered chemical reactions, and ultrasound cavitation causes locally high temperatures and pressures, how can be considered their use on an industrial basis, where the larger scales makes difficult to control the parameters?

Please comment.

The physical process of ultrasound cavitation is that the ultrasound wave propagates longitudinally in the liquid, and its alternating pressure is periodically stretched and compressed in the liquid. Due to the continuous compression and rarefaction cycle, the cavitation bubbles vary with the frequency of the sound wave pulse, and this phenomenon is called “stable cavitation”. However, the bubbles keep on growing until it reaches its critical value, high temperature (5,000 K) and high pressure (100 MPa) on the cavitation zone will be generated. This type of cavitation, called “transient cavitation”, produces shear forces and turbulence at the moment of collapse. The literature indicated that the pressure, temperature or volume can not affect the macroscopic system due to the small size of these parameters, but these parameters may affect the cell structure and increase the mass transfer process.

For detail study you can read review article:

Wen, C., Zhang, J., Zhang, H., Dzah, C. S., Zandile, M., Duan, Y., ... & Luo, X. (2018). Advances in ultrasound assisted extraction of bioactive compounds from cash crops–A review. Ultrasonics sonochemistry, 48, 538-549.

  1. Results and discussion

The structure of this section does not seem appropriate as 3.4 (β-carotene), 3.5 (Lutein) and 3.6 (Lycopene) should be subsections of 3.3 - Influence of independent variables on the extraction of different carotenoids.

Line 266, 287, 309: The comment followed. We have corrected section numbers according to your valuable suggestion in the revised manuscript.

Additionally, it is not clear why the response surface plots of the Total Carotenoids and β-carotene are grouped in a same figure, which difficult the discussion. The same happens with figure 3 that groups the response surface plots for Lutein and Lycopene.

The comment followed. We have separated all figures. “Figure 1. The response surface plots of the total carotenoids as influenced by independent variables during extraction. Figure 2. The response surface plots of the β-carotene as influenced by independent variables during extraction. Figure 4. The response surface plots of the lutein as influenced by independent variables during extraction. Figure 5. The response surface plots of the lycopene as influenced by independent variables during extraction”.

Line 361 – “…lutein and lycopene from the carrot pomace. Because it could greatly…

Please reformulate

Line 349: The comment followed. “ultrasound was a productive method for the extraction of carotenoids from the carrot pomace”

Line 365 – “…The two most productive optimum extraction conditions were obtained…”

Authors must choose: “Most productive” or “optimum”.

I am confused. Which are the “two most productive” conditions?

Line 354-355: The comment followed. The line is reformulated in the revised manuscript. “The optimum extraction conditions were obtained for the total carotenoids, β-carotene, lutein and lycopene.”

Line 367 – “In conclusion, the ultrasound-assisted extraction may successfully be employed in the industry for the extraction of carotenoids from the carrot pomace.”

Probably this sentence is overambitious as this is a laboratorial study and not a pilot plant study.

Line 355: The comment followed. The line is reformulated in the revised manuscript. “In conclusion, the ultrasound-assisted extraction has a potential for the extraction of carotenoids from the carrot pomace.”

Reviewer 3 Report

 The authors of the manuscript present a study in which an experimental design has been applied for the optimization of an analytical method to determine carotenoids in carrot pomace.

The manuscript is clear, well-written and organized. However, I miss the application of the optimized method to real samples. Thus, I recommend the publication of this study in Molecules after a major revision.

Some inputs to consider:

  1. Title: I will replace ‘valuable compounds’ by ‘carotenoids, and ‘food waste’ by ‘carrot pomace’ to be more specfic’
  2. Line 26: ‘…for total carotenoids determination, and…’
  3. Line 29: ‘more productive’ than what?
  4. Lines 86 and 92: RSM and CCD have not been defined in the Introduction.
  5. Line 92: Please, specify the three carotenoids.
  6. Line 113: Please, replace ‘hrs’ by ‘h’
  7. Lines 119-120: Please, include the extraction conditions so that the reader can carry out the extraction correctly following this protocol
  8. Line 132: include a sentence indicating that the method for determining the analytes is detailed below.
  9. Line 164: Replace ‘Analysis’ by ‘Determination’
  10. Section 2.6: Since a chromatogram is not presented, please indicate the retention time of the two analytes.
  11. Section 2.7: This analyte could not have also been determined by chromatography, together with the other two??
  12. Figure 1: I suggest dividing it in two, so that one is for the total content of carotenoids, and another for the carotene.
  13. Conclusion: A deeper comparison with previous methods for the same purposes should be included.
  14. As previously said, I strongly miss the application of the method to real samples. It should be included in the revision.

Author Response

List of Changes

Manuscript ID: Molecules-1438414

Title: “Ultrasound-assisted extraction of carotenoids from carrot pomace and their optimization through response surface methodology”

Journal: Molecules

Referee #3

Some inputs to consider:

  1. Title: I will replace ‘valuable compounds’ by ‘carotenoids, and ‘food waste’ by ‘carrot pomace’ to be more specfic’

Line 1: The comment followed and title has been changed according to your valuable comment. “Ultrasound-assisted extraction of carotenoids from carrot pomace and their optimization through response surface methodology”.

  1. Line 26: ‘…for total carotenoids determination, and…’

Line 80: It was mistakenly added. We have removed it in the revised manuscript.

  1. Line 29: ‘more productive’ than what?

Line 31-32: “more productive process than conventional techniques”. It has been added in the revised manuscript.

  1. Lines 86 and 92: RSM and CCD have not been defined in the Introduction.

Because RSM is well known design that’s why I have discussed a little bit “RSM is a generally used statistical tool for process optimization that can reduce cost, resources and time”, I have already discussed in the manuscript with reference (Line 94-99).

  1. Line 92: Please, specify the three carotenoids.

Line 104-105: The comment followed. “during ultrasonic extraction of carotenoids (β-carotene, lutein and lycopene)”.

  1. Line 113: Please, replace ‘hrs’ by ‘h’

Line 124: The comment followed. We have changed hrs with h in the revised manuscript.

  1. Lines 119-120: Please, include the extraction conditions so that the reader can carry out the extraction correctly following this protocol

Because extraction conditions were different for different run. That’s why I have already mentioned detail conditions in Table 2.

  1. Line 132: include a sentence indicating that the method for determining the analytes is detailed below.

Line 141-142: The comment followed.” For the determination of total carotenoids, lycopene, β-carotene and lutein (method for determining the analytes is detailed below)”

  1. Line 164: Replace ‘Analysis’ by ‘Determination’

Line 169: The comment followed. We have replaced analysis with determination in the revised manuscript.

  1. Section 2.6: Since a chromatogram is not presented, please indicate the retention time of the two analytes.

Line 183-184: The comment followed. We have added rentention time of the two analytes. “The β-carotene & lutein were detected at 450 nm wavelength with retention time 5.614 and 12.395 minutes respectively”

  1. Section 2.7: This analyte could not have also been determined by chromatography, together with the other two??

Because for lycopene we have already developed a method in our lab. And we have already published the method in our previous study (Jabbar, S., Abid, M., Hu, B., Wu, T., Hashim, M. M., Lei, S., & Zeng, X. (2014). Quality of carrot juice as influenced by blanching and sonication treatments. LWT-Food Science and Technology, 55, 16-21). That’s why we use the same method for the current study.

  1. Figure 1: I suggest dividing it in two, so that one is for the total content of carotenoids, and another for the carotene.

The comment followed. We have separated all figures. “Figure 1. The response surface plots of the total carotenoids as influenced by independent variables during extraction. Figure 2. The response surface plots of the β-carotene as influenced by independent variables during extraction. Figure 4. The response surface plots of the lutein as influenced by independent variables during extraction. Figure 5. The response surface plots of the lycopene as influenced by independent variables during extraction”.

  1. Conclusion: A deeper comparison with previous methods for the same purposes should be included.

In the current study, we have only optimized sonication conditions for maximum extraction of carotenoids. In the future we will work on comparison with other methods. Thanks for your valuable comment.

  1. As previously said, I strongly miss the application of the method to real samples. It should be included in the revision.

After designing 17 runs from RSM software (Design Expert) as shown in Table 2, We practically performed 17 runs in the lab. After that we added lab data into the RSM software and then optimized the conditions by using RSM software. At the end, we performed an experiment with optimized conditions in the lab and compared our results with the predicted values as shown in Table 4.

Round 2

Reviewer 1 Report

Dear Authors, 

Most of my comments have been ignored during the revision:

Please address the following comments: 

  1. Why dosage of the food waste and parameters related to sonication (intensity, frequency, power) itself were not optimized? They may affect the extraction yield. Please include. Please do not give the reference of previous studies, as different compounds were extracted there. 
  2. Include the optimization of solvent type. It is not hard to try some other solvents. 
  3. Please compare the efficiency of the UAE with the traditional and other advanced methods reported in the literature for extraction of selected compounds from carrot waste. Present in form of the table. This comment needs you to compare the published methods with this work. Present it in a Table. What is difficulty in comparing the extraction performance of this wok with the published one? 

Author Response

Referee #1

  1. Why dosage of the food waste and parameters related to sonication (intensity, frequency, power) itself were not optimized? They may affect the extraction yield. Please include. Please do not give the reference of previous studies, as different compounds were extracted there. 

Yes, its effect the extraction efficiency. That’s why in our trail experiments we got the maximum extraction yield at the mentioned conditions (Power: 750W, frequency: 20 kHz, amplitude level: 70%, ultrasonic intensity: 48 W cm-2, sample-to-solvent ratio: 1g/50mL, Probe size: 0.5 inch by keeping 2 cm its depth in the sample) and these conditions are already mentioned in section 2.4. Ultrasound-assisted extraction (UAE). Already, our lab has published 11 papers with the same sonication conditions. So, the current study is the further extension of our previous work. In the current study, our main objective was not to optimize the sonication parameters, but only on the optimization of extraction time, temperature and ethanol concentration by using response surface methodology. We hope reviewer will understand our point from the above answer.

  1. Include the optimization of solvent type. It is not hard to try some other solvents. 

The comment followed and we have added data on different solvent types in the revised manuscript.

2.3. Optimization of different solvents

Sonication technique was employed in order to extract carotenoids from carrot pomace powder powder by using the method of Jabbar et al. (2015). Preliminary studies were conducted on different solvents to determine optimum solvent concentration (25%, 50%, 75%), sample-to-solvent ratio (1g/30mL, 1g/50mL and 1g/70mL) at a temperature (40 °C). After preliminary studies, five different solvents were used, including ethanol, methanol, acetone, acetonitirile and n-hexane at 50% concentrations with 1g/50mL sample-to-solvent ratio at a temperature (40 °C). One gram sample of carrot pomace powder for each solvent was taken in 100 ml jacketed vessels separately in triplicate and 50 ml of solvent was added at 50% concentration. Ultrasonic processor of 750W (VC 750, Sonics and Materials Inc., Newtown, CT, USA) was used in this procedure (detailed procedure is mentioned in section 2.4). The yields (%) of the carotenoid extracts from carrot pomace powder by using different solvents were determined by the following formula:

The results of preliminary studies showed that ethanol showed the maximum yield of carotenoids  compared to other solvents as shown in Fig. 6. The variations in yield extractions using different solvents at different concentrations could be because of dissimilar polarities of the solvents used (Tow et al. 2011).

  1. Please compare the efficiency of the UAE with the traditional and other advanced methods reported in the literature for extraction of selected compounds from carrot waste. Present in form of the table. This comment needs you to compare the published methods with this work. Present it in a Table. What is the difficulty in comparing the extraction performance of this work with the published one? 

The comment followed. Actually, reviewer comment is related to review articles and mostly in research papers we do not add comparison in table form. But for reviewer satisfaction, we have added comparison in the table form in the revised manuscript.

Table: Application of different extraction technologies in the recovery of carotenoids

Extraction Method

Waste Material

Compounds Recovered

Conclusion

References

Water-induced Hydrocolloidal

Complexation for Extraction

Carrot Peel

β‑carotene

The adaptability of the carotene-pectin hydrocolloidal complexation in the extraction of carotene from carrot peel waste was proven to be successful. The complexation process requires no organic solvent and it relies on water addition to induce the formation of hydrocolloidal system. The purity of b-carotene fractionated from the complex is identical to the b-carotene extracted using solvent

extraction, which was 96%.

Jayesree et al., (2021)

Microwave Assisted

Extraction

Carrot Pomace & Peel

Total Carotenoids, β-carotene content

A 77.48% recovery of carotenoid was achieved successfully at optimum conditions (165 W of microwave power, 9.39 min of extraction time and 8.06:1 g/g of oil to waste ratio); hence the carotenoid extraction by using oil under microwave irradiation is a promising and efficient process for both waste uses and enrichment of oil.

The use of intermittent microwave radiation to enhance the MAE of b-carotene and carotenoids from carrot peels was investigated. Combined use of lower microwave power (180 W) and solvent volume (75 mL) or higher microwave power (300 W) and solvent volume (150 mL) along with a lower intermittency ratio (a = 1/4) resulted in higher contents of b-carotene and total carotenoids of the extracts

Elik et al., (2020);

Hiranvarachat, & Devahastin (2014)

Electrohydrodynamic‐Ultrasonic Procedure

for Extraction

Carrot Pomace

β‑carotene

In this research, the influence of the EHD process before the ultrasonic process for β-carotene extraction from carrot pomace powder was investigated. The results showed that increasing the EHD time from 2.5 to 20 min increased the β-carotene concentration.

Salehi & Dinani (2020)

Ultrasound Treatment

Carrot Slice

Total Carotenoids

The changes in carrot tissue caused by ultrasound treatment had an impact on total carotenoid content and colour changing. Ultrasonic treatment, especially in the case of using ultrasound at 35 kHz, resulted in a substantial increase of carotenoids content in comparison to raw carrot, what was probably related to the destruction of the original cellular structure and could facilitate the extraction of these compounds.

Nowacka & Wedzik (2016)

Supercritical CO2 Extraction Process

Carrot Peel

Total Carotenoids

This work aimed to assess and optimise the extraction of carotenoids from carrot peels by supercritical CO2 (SCO2), utilising ethanol as co-solvent. The evaluated variables were temperature, pressure and co-solvent concentration. According to the validated model, the optimal conditions for maximum mass yield (5.31%, d.b.) were found at 58.5 °C, 306 bar and 14.3% of ethanol, and at 59.0 °C, 349 bar and 15.5% ethanol for carotenoid recovery (86.1%).

de Andrade Lima et al., (2018)

Expeller Extraction via Lecithin-Linkers Microemulsions

Carrot Pomace

β‑carotene

This work introduces a new green extraction solvent, based on fully dilutable lecithin-linkers microemulsions (LLMs), that is used in a continuous expeller to recover β-carotene from carrot pomace obtained after carrot juice production. The results suggest that the combination of expeller and LLMs extraction have 3-6 fold increase in β-carotene extraction as compared to other extraction methods.

Lin

(2018)

Pulsed Electric Field

Carrot Puree

Carrot Pomace

Tomato Peel

Total Carotenoids, β‑carotene, Lycopene

This study shows the feasibility of using PEF treatment to develop functional natural food ingredients, for example carrot pomace with improved carotenoid extractability. Electroporation due to PEF treatment can be used to improve the extractability of carotenoids in carrot pomace with limited loss of carotenoids into the juice during extraction.

The suitable extraction conditions were obtained at extraction time 49.4 min, extraction temperature 52.2 °C and extraction ratio 1:70 (w/w). Under these conditions, the response variables were predicted to be 19.6 μg/g, 0.27 and 74 nm for β-carotene content.

The results of this work demonstrated that the application of PEF pre-treatment of moderate intensity (5 kV/cm) and relatively low energy input (5 kJ/kg) before solvent extraction process with either acetone or ethyl lactate, can represent a sustainable, environmental friendly and food safety approach to intensify the extractability of carotenoids, especially lycopene, from industrial tomato peels residues.

Pataro et al., (2020);

Roohinejad et al., (2014);

Roohinejad et al., (2014)

References:

Jayesree, N., Hang, P. K., Priyangaa, A., Krishnamurthy, N. P., Ramanan, R. N., Turki, M. A.,  & Ooi, C. W. (2021). Valorisation of carrot peel waste by water-induced hydrocolloidal complexation for extraction of carote and pectin. Chemosphere, 272, 129919.

Elik, A., Yanık, D. K., & Göğüş, F. (2020). Microwave-assisted extraction of carotenoids from carrot juice processing waste using flaxseed oil as a solvent. LWT - Food Science and Technology, 123, 109100.

Hiranvarachat, B., & Devahastin, S. (2014). Enhancement of microwave-assisted extraction via intermittent radiation: Extraction of carotenoids from carrot peels. Journal of Food Engineering, 126, 17-26.

Salehi, L., & Dinani, S. T. (2020). Application of electrohydrodynamic‐ultrasonic procedure for extraction of β-carotene from carrot pomace. Journal of Food Measurement and Characterization, 14, 3031-3039.

Nowacka, M., & Wedzik, M. (2016). Effect of ultrasound treatment on microstructure, colour and carotenoid content in fresh and dried carrot tissue. Applied Acoustics, 103, 163-171.

de Andrade Lima, M., Charalampopoulos, D., & Chatzifragkou, A. (2018). Optimisation and modelling of supercritical CO2 extraction process of carotenoids from carrot peels. The Journal of Supercritical Fluids, 133, 94-102.

Lin, S. (2018). Development of an Expeller Extraction for β-carotene from Carrot Pomace via Lecithin-linkers Microemulsions (Doctoral dissertation, University of Toronto (Canada).

Roohinejad, S., Everett, D. W., & Oey, I. (2014). Effect of pulsed electric field processing on carotenoid extractability of carrot purée. International Journal of Food Science & Technology, 49, 2120-2127.

Roohinejad, S., Oey, I., Everett, D. W., & Niven, B. E. (2014). Evaluating the effectiveness of β-carotene extraction from pulsed electric field-treated carrot pomace using oil-in-water microemulsion. Food and Bioprocess Technology, 7, 3336-3348.

Pataro, G., Carullo, D., Falcone, M., & Ferrari, G. (2020). Recovery of lycopene from industrially derived tomato processing by-products by pulsed electric fields-assisted extraction. Innovative Food Science & Emerging Technologies, 63, 102369.

Reviewer 3 Report

The authors have responded and / or justified the comments made by the reviewer in some way, and therefore I recommend the publication of the manuscript. Only a couple of minor changes are necessary:

1) Line 214: Please, replace 'minutes' by 'min'

2) Each figure must have its own sub-figures. In this sense, please replace Fig. 2D, 2E and 2F by Fig. 2A, 2B, and 2C, respectively. The same in the case of Fig. 5.

Author Response

Referee #3

The authors have responded and / or justified the comments made by the reviewer in some way, and therefore I recommend the publication of the manuscript. Only a couple of minor changes are necessary:

1) Line 214: Please, replace 'minutes' by 'min'

The comment followed and minutes are replaced with min in the revised manuscript. 

2) Each figure must have its own sub-figures. In this sense, please replace Fig. 2D, 2E and 2F by Fig. 2A, 2B, and 2C, respectively. The same in the case of Fig. 5.

The comment followed and we have changed alphabets according to your suggestion in the revised manuscript.
